# The Oral Health Situation of 12-Year-Old School Children in the Rural Region of Ilembula in Southwestern Tanzania: A Cross-Sectional Study

**DOI:** 10.3390/ijerph182212237

**Published:** 2021-11-22

**Authors:** Lisa Zumpe, Tobias Bensel, Andreas Wienke, Matilda Mtaya-Mlangwa, Jeremias Hey

**Affiliations:** 1Clinic for Prosthodontics, Martin-Luther-University, 06112 Halle, Germany; lisa.zumpe@posteo.de (L.Z.); tobias.bensel@googlemail.com (T.B.); 2Institute of Medical Epidemiology, Biostatistics, and Computer Science, Martin-Luther-University, 06112 Halle, Germany; andreas.wienke@uk-halle.de; 3Department of Preventive and Community Dentistry, Muhimbili University of Health and Allied Science, Dar es Salaam P.O. Box 65014, Tanzania; matildamtaya@yahoo.com; 4Department of Prosthodontics, Geriatric Dentistry and Craniomandibular Disorders, Charité—Universitätsmedizin Berlin, Corporate Member of Freie Universität Berlin and Humboldt Universität zu Berlin, 14197 Berlin, Germany

**Keywords:** Tanzania, DMFT, SaC, primary schools, risk indicators for caries, 12-year-olds

## Abstract

There has been no research on the prevalence of and factors associated with dental caries in rural southwestern Tanzania among schoolchildren. Determining the prevalence of and factors associated with dental caries will help to assess the need for dental intervention and prophylactic measures among children in the region. In February 2020, a cross-sectional study was conducted in the Wanging’ombe District of the Ilembula Ward. The data were collected through clinical examinations and personal interviews at two primary schools. Univariable and multivariable logistic regression analyses were performed to identify potential risk indicators for caries. The study included 319 students aged 11–12 years (average 11.92 ± 0.27 years). The mean Decayed, Missing and Filled Teeth index was 0.24 ± 0.68, and the mean Specific affected Caries Index was 1.66 ± 0.9. The greatest influences on the caries risk were poor oral hygiene (OR 8.05, 95% CI 0.49–133.23), low tooth brushing frequency (OR 3.03, 95% CI 1.26–7.26) and low level of education in parents (OR 2.63, 95% CI 0.99–6.98). Dental caries was low among students in rural areas in the Wanging’ombe District.

## 1. Introduction

An estimated 2.4 billion people suffer from decayed permanent teeth worldwide. The World Health Organization (WHO) expects an increasing prevalence for low- and middle-income countries [1]. Studies from emerging and developing countries, such as Gabon, Cambodia and Bolivia, report Decayed, Missing and Filled Teeth (DMFT) values above 4.4 in 12-year-olds. However, the generalization that developing countries generally have high DMFT values in 12-year-olds does not seem to be correct. For example, a DMFT of 0.3 among 12-year-olds in developing Tanzania has been described compared to a value of 1.7 in economically strong Norway [2].

Tanzania is a poor country. In 2020, it ranked 163 out of 189 on the Human Development Index [3]. Due to high economic growth in recent years, Tanzania nevertheless achieved formal ascent from a low-income country to a lower-middle-income country in July 2020 [4].

With a density of 0.075 dentists per 10,000 inhabitants, Tanzania ranks last in the world. Thus, it falls well short of the WHO target (1 per 7500 inhabitants = 1.3 per 10,000) [5]. A differentiated view of the dentist/inhabitant ratio with regard to rural and urban areas does not exist. A lower density can be assumed in non-urban areas. A 2014 survey of 100 dental students at Muhimbili University of Health and Allied Sciences indicated that 83% aspired to work in an urban area after graduation [6]. 

The state-administered health services in Tanzania are under the administration of the central department for oral health. In 2002 the ministry published a guideline to improve oral health care in the country. In addition to the implementation of oral health information in schools, the guideline also includes the promotion of nationwide oral health studies and the revision of existing curricula for the training of dental staff [7]. The most common dental therapy in Tanzania and offered in all dental facilities is tooth extraction [8,9,10]. The costs for a tooth extraction correspond to four times the average daily financial resources of a single person. Restorations like fillings are even nine to ten times as high [8]. In addition to the high costs of dental treatments, a lack of therapy offers is characteristic of dental care in Tanzania. Removable dentures are offered in only 32% and root canal treatments in 46% of dental facilities. The most frequent cause of the lack of therapy offers is cited as a lack of necessary expendable items and functional dental equipment [8]. The main reason for going to a dental clinic is acute toothache, with only one in four people with orofacial pain in rural areas having access to dental care facilities [9]. The reasons why dental facilities are not consulted or consulted very late were cited by those surveyed as long distances and high transport costs to the dental clinics [9]. In addition to conventional dentistry as taught at universities based on Western models, traditional African dentistry is also used in Tanzania. According to the WHO, 60–79% of the Tanzanian population currently use traditional medicine [11]. Especially in rural areas of Tanzania and in urban slums, access to western-oriented conventional medicine is restricted [12].

According to WHO data, Tanzania does not appear to have an increased prevalence or incidence of caries, with an average DMFT of 0.3 in 12-year-olds. These data are based on surveys from the east, southeast, north and northeast regions of Tanzania and are mostly older than 10 years [13,14,15,16,17,18,19,20,21,22].

In contrast, no current data are available, in particular for the southwestern part of Tanzania. With a total population of around 4.1 million, around 8.4% of the total population of mainland Tanzania live in this region consisting of the areas of Njombe, Mbeya and Ruvuma. Therefore, the objectives of this study were to determine the prevalence of dental caries by the DMFT index and Specific affected Caries index (SaC) and to assess dental attendance, oral care practices and their relationship with dental caries among 12-year-old students in a rural region in the southwestern part of Tanzania. This age was considered to be the age of global monitoring of dental caries for international comparisons and monitoring of disease trends [23].

## 2. Materials and Methods

A cross-sectional study was conducted in the southwestern part of Tanzania, more precisely in Wanging’ombe. In this district, agriculture generates almost 90% of the incomes of inhabitants. This study employed a purposive sampling design. The Wanging’ombe District is administratively divided into 21 wards. A convenience sampling was used to select one ward (Ilembula). Dental care for the population of Ilembula is covered by two dental therapists who are employed in the local hospital and five traditional healers. Dentists, on the other hand, do not practice in Ilembula, so that a dental consultation requires about an hour’s travel to the nearest town of Makambako. The total list of all primary schools found (nine) in the selected ward were then obtained, and two schools were randomly selected by a lot procedure. The total sample size was allocated to the selected schools proportional to the number of 12-year-old students present. Finally, using a list of all names of 12-year-old students as a sampling frame, study participants were randomly selected class by class in alphabetical order alternately ascending and descending. The study was conducted in February 2020 within four weeks.

The total sample size was calculated based on the primary outcome of variable dental caries and using the following assumptions: the proportion of children with dental caries was estimated as unlikely to exceed 25% based on former surveys in urban areas in Tanzania, 95% confidence interval (CI) and degree of precision 5% (distance from proportion estimate to confidence interval margin). This resulted in a sample size of 288 [24]. Assuming a non-response rate or recording error of 10% [24], the final sample was adjusted using the total 12-year-old student population in the two schools, yielding a final sample size of at least 317 students. 

Data collection was performed by a single team consisting of an examiner, a dentist, and a recording assistant who was sufficiently trained in theory and practice before the commencement of the survey. Ethical clearance for conducting the study was granted by the National Institute for Medical Research in Tanzania (NIMR/HQ/R.8a/Vol.IX/3192). Written consent was secured from the director of each school and from the parent or legal guardian of each child that participated in the study. After a brief explanation of the purpose of the study, verbal consent was obtained from the study participants. Confidentiality was assured by excluding personal identifiers and the right to withdraw at any stage of the study. 

The data were collected through clinical examinations and personal interviews at primary schools. Instruments used during data collection were sterilized before dental examination and at the end of the data collection period in Ilembula Community Hospital. Disposable gloves and masks were used during the data collection. Pupils were examined for dental caries under daylight and headlamps using a dental mirror and, if necessary, a probe. Only manifest caries lesions were recorded. In case of doubt, caries was not recorded as present. The severity of dental caries was recorded using DMFT and SaC scores. The SaC was introduced for the risk groups in populations with a low prevalence of caries. The SaC describes the mean caries experience (DMFT) in the group presenting with caries experience (DMFT > 0) [25]. Oral hygiene status was assessed using the Simplified Oral Hygiene Index (OHI-S) [26]. The OHI-S includes the presence of Debris (DI-S) and Calculus (CI-S). The labial surfaces of teeth 16, 11, 26 and 31 as well as the lingual surfaces of teeth 36 and 46 were included in the assessment. If the first molars were not present, the respective molar distal to it was assessed. If teeth 11 or 31 were missing, the next available distal tooth was included in the calculation. The assessment was carried out using a dental mirror and a probe on a scale from 0 to 3 (0 = no debris/calculus visible, 1 = up to 1/3 of the tooth surface covered with debris/calculus, 2 = 1/3 to 2/3 of the tooth surface covered with debris/calculus, 3 = more than 2/3 of the tooth surface covered with debris/calculus). For the calculation of the indices DI-S and CI-S, the individual values were added and divided by the number of areas assessed. This resulted in values between 0 and 3. The sum of DI-S and CI-S resulted in OHI-S. The OHI-S therefore assumed values between 0 and 6. During comparison with the literature, OHI-S scores were dichotomized into good oral hygiene (OHI-S < 1) and poor oral hygiene (OHI-S ≥ 1) [27]. During regression analysis, OHI-S scores were graded into low (OHI-S ≤ 1.2), medium (>1.2 to ≤3.0) and high oral hygiene indices (>3.0 to ≤6.0) according to Wei and Lang [28].

The interviews were conducted in classrooms in the local language and included the following items. Students were asked about the levels of education of their parents. For the evaluation, the educational level of the parent with the highest degree was used. Parental level of education was originally ranked from not attended school (0) to > college or university (4). For comparison, parental level of education was recoded as low education (including original categories 0, 1, and 2) and high education (including original categories 3 and 4).

Home standard was investigated with regard to the electricity and water supply, as well as the existence of a flushing toilet. 

Dental attendance was measured by the response to the question “Have you ever attended a dentist/dental therapist for treatment? Why did you go to the dentist?”. 

Sugar intake was also examined. It was distinguished by manufactured sweets, sweetened tea and sugary soft drinks. In addition, the frequency was questioned—more than once a day, once a day, seldom or never.

The oral care practices included the type of oral hygiene aid used, method of oral hygiene, and toothbrushing frequency. Frequency was graded as more than once a day, once a day, seldom or never. In addition, the tools utilized were surveyed. 

Oral health knowledge was elicited through the following statements: “Brushing your teeth prevents tooth damage. Sugar can damage teeth. Everyone has carious teeth and that’s normal. I can do nothing to prevent carious teeth. It is enough to clean the teeth with fingers and/or to rinse them with water”. The response to each question was either “yes” or “no”.

Data analysis was performed by applying descriptive statistics and regression analysis to determine factors that had an influence on whether a child belong to the caries risk group. For this purpose, the examined students were dichotomously divided into “high caries risk” when DMFT > 0 and “low caries risk” when DMFT = 0. Group comparisons were performed by means of the Mann-Whitney-U test. Potential risk indicators were first subjected individually to univariable logistic regression analysis. Variables that showed an association with caries risk were then included in a multivariable regression analysis. Based on the determined odds ratios (ORs), statements could be made about the factor by which the chance of an examined child belonging to the “high caries risk” group differs from the respective “low caries risk“ group. Because the present study was descriptive-exploratory, both *p*-values and 95% CIs were interpreted exploratively [29].

To assess the reliability of the data obtained duplicate clinical examinations were carried out with 36 randomly selected participants 3 weeks apart. Intra-examiner reliability per Cohen’s kappa value was 94% for DMFT.

## 3. Results

Out of the 319 participants included in the survey, 163 (51.1%) were females. The average age was 11.92 ± 0.27 years. The majority of the respondents (86.8%) were Christian, and the Bena ethnic group comprised 86% of the study population. 

### 3.1. DMFT Score

With DMFT = 0, over 85% of all examined students had no experience of caries and thus had primarily healthy permanent teeth. In 15% of the children examined, a DMFT ≥ 1 and thus experience of caries on the permanent teeth could be determined. This caries polarization shows that in the permanent set of teeth 15% of the students had 100% caries. (Table 1). Males had a higher SaC (1.95 ± 1.05) than females (1.46 ± 0.73) (*p* = 0.109). None of the children examined had fillings on permanent teeth. Almost 90% of the DMFT resulted from carious teeth and 10% from teeth missing due to caries. The largest DMFT value observed was 4.

### 3.2. OHI-S Parameters

Soft and/or solid plaques or calculi were found in 91.5% of the pupils. The mean OHI-S was 1.16 ± 0.72 (Table 2). One-third (34.1%) of the pupils had good oral hygiene, while two-thirds (65.9%) of the pupils had poor oral hygiene. This was composed of soft deposits in 88.8% of the pupils.

### 3.3. DMFT and OHI-S as a Function of Parental Level of Education

In one-fifth of the pupils (20.9%), the parents have not attended school. Nearly half of the pupils (49.9%) had parents with a secondary school diploma or higher. Pupils whose parents did not attend school had the highest mean DMFT, 0.42 (±0.90). Pupils whose parents had the highest possible school-leaving qualification had the lowest mean DMFT, 0.15 (±0.59). Pupils whose parents did not attend school had the highest mean OHI-S, 1.31 (±0.67). Pupils whose parents had the highest possible school-leaving qualification had the lowest mean OHI-S, 1.07 (±0.72) (Table 3).

### 3.4. Home Standard

In all, 15% of the pupils lived in a household without a domestic water supply; 65.5% of the pupils did not have a private flushing toilet supplied with tap water; and the households of 28.2% of the pupils did not have access to electricity.

### 3.5. Dental Attendence

Almost one-third (31.8%) of the children surveyed stated that they had already received dental treatment. Around 87% of these children cited pain as the reason for their respective consultations. Less than 13% of these children stated that they had been seen for a check-up.

### 3.6. Sugar Intake

Regarding the consumption of manufactured sweets, 35.7% of the pupils reported eating them more than once a day, 40.8% of the pupils reported eating them once a day, and 23.5% of the pupils reported eating them less than once a day. Sweetened tea was consumed by 16.3% of the pupils more than once a day, 60.5% of the pupils once a day, and 23.2% of the pupils less than once a day. Sugary soft drinks were consumed by 9.4% of the pupils at least twice a day, 59.3% of the pupils once a day and 31.4% of the pupils did not consume them daily (Table 4). None of the sugar intake variables showed a significant influence in the univariable logistic regression analysis (*p* > 0.05). They were no longer used for the multivariable regression analysis.

### 3.7. Oral Care Practice 

87.9% of the study participants stated that they brush their teeth at least once a day, 12.1% do not brush their teeth daily. The most commonly used tools were brushes, which 94.9% reported using. Two-thirds (67.1%) of the study participants reported using toothpaste when brushing their teeth. The traditional miswak was used by 5% of study participants. Children who reported brushing their teeth at least once or more a day had a DMFT of 0.20 (±0.51) and 0.22 (±0.71), respectively. Children who reported not brushing their teeth daily had a mean DMFT of 0.55 (±1.02), which was more than twice as high. The OHI-S, however, was similar in all groups (Table 5).

### 3.8. Oral Health Knowledge

The fact that sugar can damage teeth was known by 92.8% of respondents. The fact that simply cleaning your teeth with fingers and water is not sufficient oral hygiene was also answered correctly by 89.9% of the pupils. However, 36.9% of the pupils were not aware that brushing teeth can prevent tooth damage and 27.6% assumed that tooth damage cannot be prevented in general. Overall, 68.7% of the pupils were able to answer all questions correctly and thus demonstrated good oral health knowledge. In contrast, 31.3% were found to have poor oral health knowledge, with two or more incorrect answers (Table 6). There was no significant difference between students with a DMFT = 0 and students with a DMFT > 0 (*p* = 0.587).

### 3.9. Relationship between Different Variables and Caries Risk

Caries experience on the permanent teeth (DMFT > 0) and thus an increased caries risk was found in 47 examined pupils (14.7%). To evaluate the strength of influence of different factors on caries risk, the outcome variable caries risk was dichotomized (“high caries risk” (DMFT > 0) vs. “low caries risk” (DMFT = 0)), and a logistic regression analysis was performed (Table 7). According to the results, the greatest influences on the caries risk were oral hygiene, toothbrushing frequency and parental level of education. The less frequently the teeth were brushed, the more plaque that was present, and the lower the educational level of the parents, the more likely the children were to belong to the high caries risk group. It was found that brushing teeth less than once a day had a stronger effect (OR 3.1; *p* = 0.02) on caries risk than brushing teeth once a day (OR 1.49; *p* = 0.302) compared to brushing teeth twice a day. In addition to toothbrushing frequency, the OHI-S also had an impact on caries risk. Thus, it was determined that an increasing OHI-S was associated with an increase in caries risk. This was reflected in an OR of 2.16 (*p* = 0.031) for an OHI-S > 1.2 and ≤ 3.0 and 4.80 (*p* = 0.293) for an OHI-S > 3.0 and ≤ 6.0. Regarding the parental level of education, an increase in caries risk could be determined with decreasing educational level. The reference group here were the children in whom at least one parent had an extended secondary school qualification and thus the highest possible school qualification in Tanzania. If the highest level of education was secondary school, the risk of caries increased by 23% (OR = 1.23; *p* = 0.703), by 89% (OR = 1.89; *p* = 0.207) if the parent had completed primary school, and by 143% (OR = 2.43; *p* = 0.092) if the parent had not completed primary school. In addition, although less strongly than for the first three factors, the absence of a domestic water supply also seemed to increase the risk of belonging to the caries risk group (OR = 1.59; *p* = 0.374). 

## 4. Discussion

With a prevalence below 15%, dental caries was a less common public health problem among 12-year-old students in the Njombe region, which is in the southwestern part of Tanzania. This is higher compared to the 7.4% prevalence reported in the Arusha region in the northern part of Tanzania for people of Maasai ethnicity, but it is similar to the 14.8% for those of non-Maasai ethnicity and to the 18.3% prevalence reported in the Lindi region in the south-eastern part of Tanzania. Furthermore, it was lower than the rates described from the Dar es Salaam region (22%) in the eastern part of Tanzania [20,21,30]. These different prevalence figures could be attributed to many factors, including sociocultural differences; study setting; dietary behaviours; and differences in knowledge, attitudes and practices regarding oral care.

The total mean DMFT of the study participants was 0.24, which is as low as in the studies done in the Lindi region (rural 0.32/urban 0.37), in the Morogoro region (0.38–0.52), in the Arusha region (0.13) and in the Dar es Salaam region (0.37–0.39) in comparison to the mean value in the world (at about 2.5) [31]. The decayed component accounted for 90% of the DMFT value, as it made a 0.22 (±0.64) contribution to the mean DMFT. Only 10% counted for missing teeth. About 35 years ago in the Singida region 24% of the children had up to 12 teeth missing due to the mutilating removal of primary canines and premolars (locally called nylon teeth), making the condition more serious than dental caries in that region at that time [19]. In our investigation, no teeth were missed according to that traditional mutilation. Since the caries distribution is generally observed to be skewed, the caries index addresses individuals with the highest caries scores. The mean SaC score in this study was 1.66. This figure is low even compared to the goal set by the WHO, which suggested that countries should limit the SaC score to be below 3 by the year 2015 [32].

Our analysis revealed that pupils with a high OHI-S were about five times more likely to develop carious permanent teeth (OR = 4.80) than pupils with a low OHI-S. Nevertheless, only 34% of the pupil had good oral hygiene. This is comparable to the value of 31.5% among the Maasai and is lower than the value of 45.9% of the students in the Arusha region and below the values of 61.9% and 68% of the students in the Dar es Salaam region [20,21,30]. Within the studies, there was an association between the quality of the oral hygiene status and the presence of caries. Consequently, this association cannot be shown for all regions and ethnic groups. However, it remains unclear at which point poor and good oral hygiene status was discriminated in the other studies.

With regard to the educational level of parents, the results indicate that pupils whose parents did not attend school were about twice as likely to develop carious permanent teeth (OR = 2.41) compared to pupils in whom at least one parent had an extended secondary school. The study by Mashoto et al. also examined educational level as a factor influencing the presence of caries [21]. Children of parents with low educational levels were more likely to have caries. In the cohort from the Lindi region, there was a similar distribution of parental educational level. Among the mothers and fathers of the pupils, 44.7% and 41.6%, respectively, had low educational levels. In the present study, there was no differentiation between parents. Almost half (50.1%) of parents had a low educational level. In contrast, parents from the Maasai population areas of Tanzania mostly had a low educational level. The rate was 95% for mothers and 92% for fathers. An association with the very low caries prevalence could not be established. The data particularly highlight the high regional differences within the country [30]. In addition, the following should be taken into account—parental educational level can be difficult to ascertain because adolescents do not know this information and are unwilling to disclose it, resulting in a high rate of nonresponse, especially in groups with lower educational levels [33].

Home standard was planned as a surrogate for social status. The selected factors turned out to be of little influence. One reason for this could be that Tanzania has achieved considerable progress in infrastructure for all classes of society in recent years. The selected factors only inadequately discriminate for social status. Nearly one-third of children reported that their homes did not have electricity. Compared with Western industrial countries, this figure seems very high. However, if you look at the development of Tanzania, it is a very encouraging figure. In recent years, the supply of electricity to the population has increased rapidly [34]. In contrast, the water supply has remained strained. However, this affects larger areas and allows conclusions to be drawn about socioeconomic status only in urban regions [35]. In the survey, only 15% of children reported not having their own water supply. This figure is very low compared to the country as a whole, but the Njombe region is considered prosperous within Tanzania.

As in the Lindi region, where no dental visits were reported by 89.6% of the study participants, almost two-thirds (66.7%) of the participants in the present study said that they had never visited a dentist in their life [21]. Among the pupils who had visited a dentist, more than two-thirds (87%) presented due to dental pain, which was similar to studies done in Eritrea (82%) [36] and Kenya (73.4%) [37]. Both figures illustrate the low dentist density and lack of preventive examinations in Tanzania. However, in contrast to Eritrea, the low prevalence of caries does not make these examinations worthwhile.

Regarding the consumption of sweets, the results show changes compared to the data from 35 years ago [19]. At that time, only 13% of the students reported at least twice a day consumption of manufactured sweets, whereas in our investigation more than one- third (35.7%) of the students reported this finding. Less than a quarter of the children (23.5%) reported consuming manufactured sweets less than once a day, and 35 years ago, this figure was almost three times higher (65%) [19]. Despite this, Tanzania still has low sugar consumption. In 2018, global consumption averaged 22.6 kg, with only 9.8 kg in Tanzania, which is still below the level of sugar consumption (15 kg/person/year) at which most of the population will not get dental caries [38,39]. This could explain why sugar intake was not a risk factor in the present study. However, it should be noted that the questions regarding diet only allowed for a very rough distribution of the study cohort. In future studies, greater importance should be attached to this aspect.

Contrary to this, pupils who did brush their teeth seldom or never were about three times more likely to develop carious permanent teeth (OR = 3.10) than pupils who brushed their teeth more than once a day. More than 43% of the pupils reported that they did. This was almost 10% more than in the survey in the Lindi region and more than 14% less than in a survey conducted 35 years ago in the rural areas of Dodoma, Singida and Morogoro [19].

Almost all (94.9%) of the study participants said they used brushes. In contrast, only 5% said they used the traditional miswak. In the previous study, over 35 years ago, the proportions were reversed [19]. At that time, 90% of the children interviewed said they used chewing or brushing sticks made by the families themselves.

In the present study, students with poor oral health knowledge had higher caries experience and poorer oral hygiene. Overall, however, a good general knowledge of basic oral diseases and oral health was found. This corresponds to the results of previously published studies from northern and eastern Tanzania. Here, too, good basic oral health knowledge was found among schoolchildren between 12 and 17 years of age [17,18,24]. In Tanzania, oral health education in schools has been implemented in the country’s curriculum since 2002. However, whether comprehensive oral health knowledge has a positive impact on oral hygiene behaviour and oral health is controversial in the literature. For example, studies show that the benefits of individual and group-based oral health education are equivalent to the clinical benefits of professional teeth cleaning and can thus be considered a successful tool in oral health prophylaxis [40].

Several aspects should be considered when interpreting and comparing these results with previous studies conducted in Tanzania. First, the clinical examinations were carried out in daylight with additional headlamps. The examination of the children took place in a sitting upright position in open corridors of school buildings. This may have influenced decision making in clinical diagnosis, especially in the case of visible plaque and, to some extent, caries. Second, carious lesions were confirmed by visual inspection only; saliva could not be blown away or removed by suction. Therefore, only manifest caries lesions could be determined.

However, the present results should be interpreted in the light of limitations that include a cross-sectional design and use of self-reported measures. Since the present data rely on self-reporting, they might have been biased by under- and overreporting due to socially desirable responses. However, the core questions utilized in this study have shown good validity and reliability in previous studies focusing on children and adolescents in Tanzania.

In future studies, the focus should be more on a detailed dietary history and the questioning of previous caries experience. Furthermore, it would be of interest to include more precise information on the fluoride content of the toothpastes used. These aspects would be a useful addition to identify more precisely the causes of caries development in the region.

## 5. Conclusions

The prevalence of dental caries was low among students in rural areas in the Wanging’ombe District. In 2020, the prevalence of dental caries in Tanzania remains low. Toothbrushing frequency, parental level of education and oral care practices were the influencing factors for dental caries.

## Figures and Tables

**Table 1 ijerph-18-12237-t001:** Mean values of the Decayed, Missing and Filled Teeth (DMFT) parameters and the Specific affected Caries Index (SaC).

	DMFT	D	M	F	SaC
Total (*n* = 319)	0.24 (±0.68)	0.22 (±0.64)	0.03 (±0.21)	0.00	1.66 (±0.90)
Female (*n* = 163)	0.25 (±0.63)	0.21 (±0.58)	0.04 (±0.24)	0.00	1.46 (±0.73)
Male (*n* = 156)	0.24 (±0.73)	0.22 (±0.70)	0.01 (±0.16)	0.00	1.95 (±1.05)

**Table 2 ijerph-18-12237-t002:** Mean values of the Simplified Oral Hygiene Index (OHI-S) parameters.

	Oral Hygiene Index (Simplified)	Debris Index (Simplified)	Calculus Index (Simplified)
Total (*n* = 319)	1.16 (±0.72)	1.03 (±0.58)	0.14 (±0.33)
Female (*n* = 163)	1.05 (±0.69)	0.93 (±0.54)	0.12 (±0.32)
Male (*n* = 156)	1.28 (±0.72)	1.14 (±0.61)	0.15 (±0.35)

**Table 3 ijerph-18-12237-t003:** Mean DMFT und OHI-S ranked according to parental level of education.

	Did Not Attend School	Primary School	Secondary School	College or University
*n* (%)	65 (20.9)	91 (29.3)	82 (26.4)	73 (23.5)
DMFT	0.42 (±0.90)	0.22 (±0.53)	0.23 (±0.72)	0.15 (±0.59)
OHI-S	1.31 (±0.67)	1.17 (±0.78)	1.09 (±0.65)	1.07 (±0.72)

**Table 4 ijerph-18-12237-t004:** Frequency of sugar intake.

	More Than Once a Day	Once a Day	Seldom/Never
Manufactured sweets	114 (35.7%)	130 (40.8%)	75 (23.5%)
Sweetened tea	52 (16.3%)	193 (60.5%)	74 (23.2%)
Sugary soft drinks	30 (9.4%)	189 (59.3%)	100 (31.3%)

**Table 5 ijerph-18-12237-t005:** Mean DMFT and OHI-S in relation to the frequency of toothbrushing.

	More Than Once a Day	Once a Day	Seldom/Never
*n* (%)	137 (43.8)	138 (44.1)	38 (12.1)
DMFT	0.22 (±0.71)	0.20 (±0.51)	0.55 (±1.02)
OHI-S	1.16 (±0.70)	1.13 (±0.75)	1.28 (±0.66)

**Table 6 ijerph-18-12237-t006:** Mean DMFT and OHI-S depending on oral health knowledge.

	Good	Poor
*n* (%)	219 (68.7)	100 (31.3)
DMFT	0.24 (±0.67)	0.40 (±1.00)
OHI-S	1.03 (±0.72)	1.30 (±0.71)

**Table 7 ijerph-18-12237-t007:** Logistic regression to assess the predictive influence of different variables on caries risk.

	Univariable Regression		Multivariable Regression		
	OR ^1^	95% CI ^2^	OR ^1^	95% CI ^2^	*p*-Value
	Simplified Oral Hygiene Index (OHI-S)				
Low (<1.2)	Reference group				
Medium (1.2 ≤ 3.0)	2.06	1.09–3.89	2.16	1.1–4.33	0.031
High (>3.0)	8.05	0.49–133.23	4.8	0.26–89.61	0.293
	Parental level of education				
College/university	Reference group				
Secondary school	1.33	0.48–3.69	1.23	0.42–3.58	0.703
Primary school	1.99	0.77–5.13	1.89	0.7–5.09	0.207
Did not attend school	2.63	0.99–6.98	2.43	0.87–6.82	0.09
	Domestic water supply				
Existent	Reference group				
Non-existent	1.22	0.48–3.05	1.59	0.57–4.44	0.374
	Tooth brushing frequency				
More than once a day	Reference group				
Once a day	1.26	0.62–2.55	1.49	0.7–3.16	0.302
Seldom/never	3.03	1.26–7.26	3.1	1.2–8.02	0.02

^1^ Odds ratio (OR), ^2^ Confidence interval (CI).

## Data Availability

The data presented in this study are available on request from the corresponding author.

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
