# Peer review of "The Oral Health Situation of 12-Year-Old School Children in the Rural Region of Ilembula in Southwestern Tanzania: A Cross-Sectional Study"

_ijerph, 2021, doi:10.3390/ijerph182212237_

Round 1

Reviewer 1 Report

The introductions could be written a little better with some English changes

Methodology:

The school and study participants were randomly selected-how was randomisation done?

How long was the study conducted for?

Data collection was done by a dental medicine-not sure about this role.

Results:

The methodology says 317 samples were yielded but results mention 319.

There is mention of one religious and ethnic group but not others

Table 1: put the whole table in one page.

Author Response

@ reviewer 1:

Thank you for reviewing our manuscript and suggesting improvements. We have made the following changes.

Reviewer 1 comment: Methodology: The school and study participants were randomly selected-how was randomisation done?

Our answer/modification: We chose simple randomization procedures and completed the two sentences accordingly as follows:

“The total list of all primary schools found (nine) in the selected ward were then obtained, and two schools were randomly selected by a lot procedure.”

“Finally, using a list of all names of 12-year-old students as a sampling frame, study participants were randomly selected class by class in alphabetical order alternately ascending and descending.”

Reviewer 1 comment: How long was the study conducted for?

Our answer/modification: We have specified the sentence as follows:

“The study was conducted in February 2020 within four weeks.”

Reviewer 1 comment: Data collection was done by a dental medicine-not sure about this role.

Our answer/modification: We have corrected our mistake and changed the word in the sentence as follows:

“Data collection was performed by a single team consisting of an examiner, a dentist, and a recording assistant who was sufficiently trained in theory and practice before the commencement of the survey.”

Reviewer 1 comment: Results: The methodology says 317 samples were yielded but results mention 319.

Our answer/modification: According to our sample size calculation, at least 317 students were required. Two more were examined, making a total of 319. We have clarified the sentence as follows:

“Assuming a non-response rate or recording error of 10% [18], the final sample was adjusted using the total 12-year-old student population in the two schools, yielding a final sample size of at least 317 students.”

Reviewer 1 comment: There is mention of one religious and ethnic group but not others.

Our answer/modification: The information is of secondary importance. We have also deleted the sentence on the recommendation of the 2nd reviewer.

Reviewer 1 comment: Table 1: put the whole table in one page.

Our answer/modification: Thank you very much for the advice. We have adjusted the formatting.

Reviewer 2 Report

Thanks for the opportunity to read and suggest some improvements to the authors of the article:  Predictors of dental caries in 12-year-old students in 2 southwestern Tanzania: a cross-sectional study

Títle.

Since it is a cross-sectional study, it would be better to use the term risk indicator and not predictor.

The title is not correlated with the objectives of the study and conclusion.

the objectives of this study were to determine the prevalence of dental caries by the DMFT index and Specific affected Caries index (SaC) and to assess dental attendance, oral care practices and their relationship with dental caries among 12- year-old students in a rural region in the southwestern part of Tanzania.

Results

It is suggested to delete: "The majority of the respondents (86.8%) were 135 Christian, and the Bena ethnic group comprised 86% of the study population."

3.1. DMFT score

“Males had a 143 higher SaC (1.95 ± 1.05) than females (1.46 ± 0.73) “. Is the difference significant?, add p.

Table 3. Mean DMFT und OHI-S ranked according to parental level of education.

Why didn't you include the SaC?

Table 5. Mean DMFT and OHI-S in relation to the frequency of toothbrushing.

Why didn't you include the SaC?

Table 6. Mean DMFT and OHI-S depending on oral health knowledge.

It does not establish whether the differences between the means are significant

Why didn't you include the SaC?

Table 3. Mean DMFT und OHI-S ranked according to parental level of education

It does not establish if the differences are significant and I suggest including the data from SaC scores.

Table 7. Logistic regression to assess the predictive influence of different variables on caries risk.

I suggest including the ORs of sugar consumption.

Discusion:

pupils with a high OHI-S were about five times more likely to 259 develop carious permanent teeth (OR = 4.80) than pupils with a low OHI-S. Nevertheless, 260 only 34% of the pupil had good oral higiene. You cannot come to this conclusion if you do not calculate the risk ratio and risk difference.

Did you consider the presence of hidden sugars in the diet?

What is the traditional drink for breakfast and dinner in this Tanzania region?

If it is made from a ground cereal such as wheat, corn, rice or sorghum, this may be more important than consuming candies, to attend or discuss some definition or conceptualization of the caries process such as:

Dental caries is a biofilm-mediated, diet modulated, multifactorial, non-communicable, dynamic disease resulting in net mineral loss of dental hard tissues [Fejerskov 1997; Pitts et al., 2017]. It is determined by biological, behavioral, psychosocial, and environmental factors.

Machiulskiene V, Campus G, Carvalho JC, Dige I, Ekstrand KR, Jablonski-Momeni A, Maltz M, Manton DJ, Martignon S, Martinez-Mier EA, Pitts NB, Schulte AG, Splieth CH, Tenuta LMA, Ferreira Zandona A, Nyvad B. Terminology of Dental Caries and Dental Caries Management: Consensus Report of a Workshop Organized by ORCA and Cariology Research Group of IADR. Caries Res. 2020;54(1):7-14.

Author Response

@ reviewer 2:

Thank you for reviewing our manuscript and suggesting improvements. We have made the following changes.

Reviewer 2 comment: Títle. Since it is a cross-sectional study, it would be better to use the term risk indicator and not predictor.

Our answer/modification: Thank you for the comment, we have removed the word predictors and chosen risk indicator instead.

Reviewer 2 comment: The title is not correlated with the objectives of the study and conclusion.

Our answer/modification: This is correct; we have changed the title as follows:

“The oral health situation of 12-year-old school children in the rural region of Ilembula in southwestern Tanzania: a cross-sectional study”

Reviewer 2 comment: Results. It is suggested to delete: "The majority of the respondents (86.8%) were 135 Christian, and the Bena ethnic group comprised 86% of the study population."

Our answer/modification: We have removed the sentence.

Reviewer 2 comment: 3.1. DMFT score “Males had a 143 higher SaC (1.95 ± 1.05) than females (1.46 ± 0.73)“. Is the difference significant?, add p.

Our answer/modification: Thank you very much for this comment. Taking into account the exploratory (and not hypothesis driven confirmatory) nature of this investigation we refer to the ongoing methodological discussion about the dichotomization of results in significant/non-significant results (e.g. Amrhein et al. (2016) Retire statistical significance. Nature 567, 305 – 307; Wasserstein et al. (2019) Moving to a World Beyond “p < 0.05” The American Statistician 73, S1, 1 – 19). Therefore, it is not useful (especially in an exploratory study) to classify results as significant/not significant. But to give the readers more detailed information, we added the p-values linked to the comparisons requested by the reviewer:

“Males had a higher SaC (1.95 ± 1.05) than females (1.46 ± 0.73) (p=0.109).”

Reviewer 2 comment: Table 3. Mean DMFT und OHI-S ranked according to parental level of education. Why didn't you include the SaC?

Our answer/modification: The SaC describes a subgroup. The SaC describes the mean caries experience (DMFT) in the group presenting caries experience (DMFT > 0). The aim of the article was to represent a cross-section of all children in the region. Both the DMFT and the SaC are comparatively low. In the study, only 15% of the children were affected by carious defects. To more accurately determine the causes of caries development, it is necessary to include a much larger study cohort. This should be considered as a goal of future studies. We have not made any changes in the text.

Reviewer 2 comment: Table 5. Mean DMFT and OHI-S in relation to the frequency of toothbrushing. Why didn't you include the SaC?

Our answer/modification: The SaC describes a subgroup. The SaC describes the mean caries experience (DMFT) in the group presenting caries experience (DMFT > 0). The aim of the article was to represent a cross-section of all children in the region. Both the DMFT and the SaC are comparatively low. In the study, only 15% of the children were affected by carious defects. To more accurately determine the causes of caries development, it is necessary to include a much larger study cohort. This should be considered as a goal of future studies. We have not made any changes in the text.

Reviewer 2 comment: Table 6. Mean DMFT and OHI-S depending on oral health knowledge. It does not establish whether the differences between the means are significant. Why didn't you include the SaC?

Our answer/modification: The SaC describes a subgroup. The SaC describes the mean caries experience (DMFT) in the group presenting caries experience (DMFT > 0). The aim of the article was to represent a cross-section of all children in the region. Both the DMFT and the SaC are comparatively low. In the study, only 15% of the children were affected by carious defects. To more accurately determine the causes of caries development, it is necessary to include a much larger study cohort. This should be considered as a goal of future studies. We have not made any changes in the text.

Reviewer 2 comment: Table 3. Mean DMFT und OHI-S ranked according to parental level of education. It does not establish if the differences are significant and I suggest including the data from SaC scores.

Our answer/modification: The SaC describes a subgroup. The SaC describes the mean caries experience (DMFT) in the group presenting caries experience (DMFT > 0). The aim of the article was to represent a cross-section of all children in the region. Both the DMFT and the SaC are comparatively low. In the study, only 15% of the children were affected by carious defects. To more accurately determine the causes of caries development, it is necessary to include a much larger study cohort. This should be considered as a goal of future studies. We have not made any changes in the text.

Reviewer 2 comment: Table 7. Logistic regression to assess the predictive influence of different variables on caries risk. I suggest including the ORs of sugar consumption.

Our answer/modification: Thank you for pointing this out. As described in the material and methods section, only the significant risk indicators were used for the multivariable regression. We have added the following sentence for clarification.

“None of the sugar intake variables showed a significant influence in the univariable logistic regression analysis (p>0.05). They were no longer used for the multivariable regression analysis.”

Discussion:

Reviewer 2 comment: pupils with a high OHI-S were about five times more likely to develop carious permanent teeth (OR = 4.80) than pupils with a low OHI-S. Nevertheless, only 34% of the pupil had good oral higiene. You cannot come to this conclusion if you do not calculate the risk ratio and risk difference. Did you consider the presence of hidden sugars in the diet? What is the traditional drink for breakfast and dinner in this Tanzania region? If it is made from a ground cereal such as wheat, corn, rice or sorghum, this may be more important than consuming candies, to attend or discuss some definition or conceptualization of the caries process.

Our answer/modification: Thank you for the advice. Our nutritional history focused too little on the regional situation. rather globally used few questions. In future studies, we should attach more importance to this aspect. We have added the following sentences to the manuscript:

“In future studies, the focus should be more on a detailed dietary history and the questioning of previous caries experience. Furthermore, it would be of interest to include more precise information on the fluoride content of the toothpastes used. These aspects would be a useful addition to identify more precisely the causes of caries development in the region.”

Reviewer 3 Report

The data about dental caries is important for WHO to monitor the global prevalence of this disease. However, it was shown that the data in Tanzania needed to be revisited for its accuracy. The study contributed to the public health data for the prevention and management of dental caries in developing countries.  

The study is well designed to address the question of research. The results and analysis are well presented and support well the aim of the study. Authors discussed well the findings, confounding factors and limitations of the study. Therefore, the topic can be of interest for the readership for
IJERPH. The manuscript was carefully written.

I suggest minor revisions:

In the introduction, page 1, L49-51: please recap the available data about predictors of caries in Tanzania. Also, mention the demographic of Southwestern. 

Discussion:

Please mention if the previous caries experience in these subjects could affect the results as a confounding factor.

Author Response

@ reviewer 3:

Thank you for reviewing our manuscript and suggesting improvements. We have made the following changes.

Reviewer 3 comment: In the introduction, page 1, L49-51: please recap the available data about predictors of caries in Tanzania. Also, mention the demographic of Southwestern. 

Our answer/modification: We completed the text as follows:

“According to WHO data, Tanzania does not appear to have an increased prevalence or incidence of caries with an average DMFT of 0.3 in 12-year-olds.”

“With a total population of around 4.1 million, around 8.4% of the total population of mainland Tanzania live in this region consisting of the areas of Njombe, Mbeya and Ruvuma.”   

Reviewer 3 comment: Discussion: Please mention if the previous caries experience in these subjects could affect the results as a confounding factor.

Our answer/modification: We did not include the caries experience. This would certainly be an interesting aspect for future studies. We have added the following section:

“In future studies, the focus should be more on a detailed dietary history and the questioning of previous caries experience. Furthermore, it would be of interest to include more precise information on the fluoride content of the toothpastes used. These aspects would be a useful addition to identify more precisely the causes of caries development in the region.”

Reviewer 4 Report

The manuscript deals with predictors for dental caries among 12-year olds in Tanzania. 319 children were included and examined in the school facilities where the children were also interviewed about for example their oral hygiene practices, intake of food items with a high sugar content and their living situation. There are several issues the authors need to clarify and address, please see below.

Title: “students” is not “children” more appropriate?

Abstract: it would be valuable for the reader if the variables registered were stated in the abstract. Were there any statistically significant differences or correlations? Please include p-values.

Line 24. “weak oral hygiene”- do you mean “poor oral hygiene”?

Please consider including figures for caries prevalence.

You write in the beginning of the abstract that it is important to assess the prevalence of dental caries and factors associated with dental caries in order to assess the need for dental interventions and prophylactic measures. You then conclude that the prevalence of dental caries was low, 15%? Please include a line about what you think about the need for dental interventions and prophylactic measures.

Introduction: please include information about whether parents get information about oral and dental health from health care or not, and also about possible other sources of information about oral and dental health like day care or school. You mention a little about this in the discussion section, but it should be mentioned here too.

Please include information about the availability of dental clinics in the area studied. Are children normally going to check-ups at the dental clinic or is it only acute treatments? Is it costly to go to the dental clinic?

Material and methods:

Line 78, “ a dental medicine”- please clarify.

Line 90. You write in the discussion that the examinations were conducted in the corridor, please give that information here. Did you have some kind of chair which could be tilted to make the examination easier? And what about time of the day for the examinations?

Did you not use a probe to find possible caries lesions? Please clarify that it was only manifest caries lesions you could detect.

OHI-S – please describe which teeth were used for scoring and how for example plaque was scored, ranges of scores.

Line 97-99. Should this not be moved to the Data analysis section?

Line 100. Please include information about where the interviews took place.

Line 114. Oral hygiene aids- what about fluoride in the toothpaste? Is that available in Tanzania and with what fluoride concentration?

Line 123. “membership in the caries risk group”- please rephrase

Results:

Please include p-values where appropriate or add that the difference was not statistically significant.

The information about duplicate examination should be moved to the MM section. It is not clear why this was done, please clarify.

Line 141. “primary healthy dentition”- please clarify.

As I understand it, it was 15% of the children who had a DMFT > 0? It would be interesting if the authors wrote more about these results.

Line 145. 90% of the DMFT was due to carious teeth- what happened with these children? Were they referred to a dental clinic?

The same data should not be presented both in text and table. – please change.

Discussion:

Please elaborate on the advantages of doing the clinical examinations in a dental clinic, for example possible to take x-rays to detect approximal caries, possible to remove dental plaque and to dry the dental surfaces to detect caries. Why wasn’t a probe used to detect carious lesions. Please also explain why the children were not taken to a dental clinic for clinical examination and interview.

You asked the children about their intake of sweets, sweetened tea and sugary soft drinks. It would be valuable for the reader to get some information about the availability of these products and also about the cost.

You write in the results section that 65.9% of the children had poor oral hygiene- Please elaborate on how this was handled. Poor oral hygiene is a risk factor not only for caries but also for periodontal diseases. Were the children given information about the importance of brushing their teeth and instructions on how to brush their teeth? What was the reason for poor oral hygiene? Did they even have a tooth-brush?

The consumption of products with a high concentration of easily fermentable carbohydrates was high. Were the children given any information or advice regarding diet? Please also consider adding some information about what other food-items an ordinary diet for Tanzanian children consists of.

Despite low dental attendance, poor oral hygiene, not using toothpaste, and a high intake of easily fermentable carbohydrates, the caries prevalence was lower than what might have been expected. What about fluoride in the drinking water? The authors should elaborate more about possible explanations to the caries prevalence.

Line 328. Please describe the oral health education in schools. What ages of the children? Who is in charge of the education?

Conclusion: it would be nice if the authors could add some suggestions on how to improve tooth-brushing frequency and reduced intake of easily fermentable carbohydrates and thereby reduce the risk of caries. The authors should also consider including a line about what you they about the need for dental interventions and prophylactic measures for these children.

Author Response

@ reviewer 4:

Thank you for reviewing our manuscript and suggesting improvements. We have made the following changes.

Reviewer 4 comment: Title: “students” is not “children” more appropriate?

Our answer/modification: We have adjusted the title as follows:

“The oral health situation of 12-year-old school children in the rural region of Ilembula in southwestern Tanzania: a cross-sectional study”

Reviewer 4 comment: Abstract: it would be valuable for the reader if the variables registered were stated in the abstract.

Our answer/modification: Thank you for the advice. The number of words would exceed the volume of the abstract section. We have therefore limited ourselves to the most important information.We therefore hope for your indulgence that we could not implement this remark.

Reviewer 4 comment: Were there any statistically significant differences or correlations? Please include p-values.

Our answer/modification: Thank you very much for this comment. Taking into account the exploratory (and not hypothesis driven confirmatory) nature of this investigation we refer to the ongoing methodological discussion about the dichotomization of results in significant/non-significant results (e.g. Amrhein et al. (2016) Retire statistical significance. Nature 567, 305 – 307; Wasserstein et al. (2019) Moving to a World Beyond “p < 0.05” The American Statistician 73, S1, 1 – 19). Therefore, it is not useful (especially in an exploratory study) to classify results as significant/not significant. But to give the readers more detailed information, we added the p-values linked to the comparisons requested by the reviewer. For better understanding, we have added the following information to the manuscript:

“Males had a higher SaC (1.95 ± 1.05) than females (1.46 ± 0.73) (p=0.109).”

“None of the sugar intake variables showed a significant influence in the univariable logistic regression analysis (p>0.05). They were no longer used for the multivariable regression analysis.”

“There was no significant difference between students with a DMFT = 0 and students with a DMFT > 0 (p=0.587).”

Reviewer 4 comment: Line 24. “weak oral hygiene”- do you mean “poor oral hygiene”?

Our answer/modification: This is correct, we have adjusted the sentence as follows:

“The greatest influences on the caries risk were poor oral hygiene (OR 8.05, 95% CI 0.49–133.23), low tooth brushing frequency (OR 3.03, 95% CI 1.26–7.26) and low level of education in parents (OR 2.63, 95% CI 0.99–6.98).”    

Reviewer 4 comment: Please consider including figures for caries prevalence.

Our answer/modification: Thank you for this comment. A figure would certainly enhance the article. However, there would be a partial duplication of information with the content from the table. More information can be placed in the table in a more precise way. We have therefore decided to keep the tables and to do without figures.

Reviewer 4 comments: You write in the beginning of the abstract that it is important to assess the prevalence of dental caries and factors associated with dental caries in order to assess the need for dental interventions and prophylactic measures. You then conclude that the prevalence of dental caries was low, 15%? Please include a line about what you think about the need for dental interventions and prophylactic measures. Introduction: please include information about whether parents get information about oral and dental health from health care or not, and also about possible other sources of information about oral and dental health like day care or school. You mention a little about this in the discussion section, but it should be mentioned here too.

Our answer/modification: Thank you for pointing this out. We have added the following sentences for clarification:

“The state-administered health services in Tanzania are under the administration of the central department for oral health. In 2002 the ministry published a guideline to improve oral health care in the country. In addition to the implementation of oral health information in schools, the guideline also includes the promotion of nationwide oral health studies and the revision of existing curricula for the training of dental staff [7]. The most common dental therapy in Tanzania and offered in all dental facilities is tooth extraction [8-10]. The costs for a tooth extraction correspond to four times the average daily financial resources of a single person. Restorations like fillings are even nine to ten times as high [8]. In addition to the high costs of dental treatments, a lack of therapy offers are characteristic of dental care in Tanzania. Removable dentures are offered in only 32% and root canal treatments in 46% of dental facilities. The most frequent cause of the lack of therapy offers is cited as a lack of necessary expendable items and functional dental equipment [8]. The main reason for going to a dental clinic is acute toothache, with only one in four people with orofacial pain in rural areas having access to dental care facilities [9]. The reasons why dental facilities are not consulted or consulted very late were cited by those surveyed as long distances and high transport costs to the dental clinics [9]. In addition to conventional dentistry as taught at universities based on Western models, traditional African dentistry is also used in Tanzania. According to the WHO, 60-79% of the Tanzanian population currently use traditional medicine [11]. Especially in rural areas of Tanzania and in urban slums, access to western-oriented conventional medicine is restricted [12].”

Reviewer 4 comments: Please include information about the availability of dental clinics in the area studied. Are children normally going to check-ups at the dental clinic or is it only acute treatments? Is it costly to go to the dental clinic?

Our answer/modification: We have added the following section for this information:

“Dental care for the population of Ilembula is covered by two dental therapists who are employed in the local hospital and five traditional healers. Dentists, on the other hand, do not practice in Ilembula, so that a dental consultation must take about an hour to the nearest town of Makambako.”

Reviewer 4 comment: Material and methods: Line 78, “ a dental medicine”- please clarify.

Our answer/modification: We have corrected our mistake and changed the word in the sentence as follows:

“Data collection was performed by a single team consisting of an examiner, a dentist, and a recording assistant who was sufficiently trained in theory and practice before the commencement of the survey.”

Reviewer 4 comment: Line 90. You write in the discussion that the examinations were conducted in the corridor, please give that information here. Did you have some kind of chair which could be tilted to make the examination easier? And what about time of the day for the examinations? Did you not use a probe to find possible caries lesions? Please clarify that it was only manifest caries lesions you could detect.

Our answer/modification: We have added and corrected the following section in Materials and Methods section for this information: “Pupils were examined for dental caries under daylight and headlamps using a dental mirror and, if necessary, a probe. Only manifest caries lesions were recorded. In case of doubt, caries was not recorded as present.”

Our answer/modification: We have added the following section in Discussion section for this information: “First, the clinical examinations were carried out in daylight with additional headlamps. The examination of the children took place in a sitting upright position in open corridors of school buildings.”

Reviewer 4 comment: OHI-S – please describe which teeth were used for scoring and how for example plaque was scored, ranges of scores.

Our answer/modification: Thank you for the comment. We have added the following paragraph to the manuscript to clarify the methodology:

Reviewer 4 comment: OHI-S – please describe which teeth were used for scoring and how for example plaque was scored, ranges of scores.

”The OHI-S includes the presence of Debris (DI-S) and Calculus (CI-S). The labial surfaces of teeth 16, 11, 26 and 31 as well as the lingual surfaces of teeth 36 and 46 were included in the assessment. If the first molars were not present, the respective molar distal to it was assessed. If teeth 11 or 31 were missing, the next available distal tooth was included in the calculation. The assessment was carried out using a dental mirror and a probe on a scale from 0 to 3 (0 = no debris / calculus visible, 1 = up to 1/3 of the tooth surface covered with debris / calculus, 2 = 1/3 to 2/3 of the tooth surface covered with debris / calculus, 3 = more than 2/3 of the tooth surface covered with debris / calculus). For the calculation of the indices DI-S and CI-S, the individual values ​​were added and divided by the number of areas assessed. This resulted in values ​​between 0 and 3. The sum of DI-S and CI-S resulted in OHI-S. The OHI-S therefore assumed values ​​between 0 and 6.”

Reviewer 4 comment: Line 97-99. Should this not be moved to the Data analysis section?

Our answer/modification: Thank you very much for the suggestion. We had decided on this classification in advance and an established procedure for it. We believe that it should remain at the position in the manuscript especially now after the more extensive description of the OHI-S and hope for your agreement.

Reviewer 4 comment: Line 100. Please include information about where the interviews took place.

Our answer/modification: We completed the text as follows:

 “The interviews were conducted in classrooms in local language and included the following items.”

Reviewer 4 comment: Line 114. Oral hygiene aids- what about fluoride in the toothpaste? Is that available in Tanzania and with what fluoride concentration?

Our answer/modification: We did not ask about the fluoride content of toothpaste in our study. It was not the subject of the investigation. We would therefore not make any changes in this respect. However, a publication on available fluoride-containing toothpastes has appeared last year: Monica Mbaraka Ndoile, Total and Available Fluoride Content in Toothpastes Sold in Dar es Salaam, Tanzania Journal of Science 46(3): 851-858, 2020, ISSN 0856-1761, e-ISSN 2507-7961” In future studies, this might be a useful parameter to include.

Reviewer 4 comment: Line 123. “membership in the caries risk group”- please rephrase

Our answer/modification: We have rephrased the sentence as follows:

“Data analysis was performed by applying descriptive statistics and regression analysis to determine factors that had an influence whether a child belong to the caries risk group.” 

Reviewer 4 comment: Results: Please include p-values where appropriate or add that the difference was not statistically significant.

Our answer/modification: Thank you very much for this comment. Taking into account the exploratory (and not hypothesis driven confirmatory) nature of this investigation we refer to the ongoing methodological discussion about the dichotomization of results in significant/non-significant results (e.g. Amrhein et al. (2016) Retire statistical significance. Nature 567, 305 – 307; Wasserstein et al. (2019) Moving to a World Beyond “p < 0.05” The American Statistician 73, S1, 1 – 19). Therefore, it is not useful (especially in an exploratory study) to classify results as significant/not significant. On the contrary, it is to be feared that the results will be overinterpreted. We would therefore refrain from specifying a significance value and would like to use it only for hypothesis-testing studies. For better understanding, we have added the following information to the manuscript:

“Males had a higher SaC (1.95 ± 1.05) than females (1.46 ± 0.73) (p=0.109).”

“None of the sugar intake variables showed a significant influence in the univariable logistic regression analysis (p>0.05). They were no longer used for the multivariable regression analysis.”

“There was no significant difference between students with a DMFT = 0 and students with a DMFT > 0 (p=0.587).”

Reviewer 4 comment: The information about duplicate examination should be moved to the MM section. It is not clear why this was done, please clarify.

Our answer/modification: We have moved the information to the materials and methods section and added as follows: “To assess the reliability of the data obtained duplicate clinical examinations were carried out with 36 randomly selected participants 3 weeks apart. Intra-examiner reliability per Cohen’s kappa value was 94% for DMFT.”

Reviewer 4 comment: Line 141. “primary healthy dentition”- please clarify. As I understand it, it was 15% of the children who had a DMFT > 0? It would be interesting if the authors wrote more about these results.

Our answer/modification: For better understanding and to avoid duplication, we have amended the section as follows: “With DMFT = 0, over 85% of all examined students had no experience of caries and thus had primarily healthy permanent teeth. In 15% of the children examined, a DMFT ≥ 1 and thus experience of caries on the permanent teeth could be determined. This caries polarization shows that in the permanent set of teeth 15% of the students had 100% of the caries.”

Reviewer 4 comment: Line 145. 90% of the DMFT was due to carious teeth- what happened with these children? Were they referred to a dental clinic?

Our answer/modification: Our vote was limited to the examination of the children. The option of treatment was not allowed. The parents, teachers and students were informed about this in advance. We informed the parents of the affected children about the examination results. We did not make any changes in the text.

Reviewer 4 comment: Discussion: Please elaborate on the advantages of doing the clinical examinations in a dental clinic, for example possible to take x-rays to detect approximal caries, possible to remove dental plaque and to dry the dental surfaces to detect caries. Please also explain why the children were not taken to a dental clinic for clinical examination and interview.

Our answer/modification: Thank you for the suggestions. The study was conducted according to WHO guidelines for dental screening. The performance of radiographs is not even permitted here. In order to keep the data comparable with other regions, also worldwide, the WHO specifications were adhered to. Screening examinations of children are usually performed at schools or similar institutions worldwide. We have not made any changes to the text.

Reviewer 4 comment: Why wasn’t a probe used to detect carious lesions.

Our answer/modification: We added in the materials end methods section:

“Pupils were examined for dental caries under daylight and headlamps using a dental mirror and, if necessary, a probe. Only manifest caries lesions were recorded. In case of doubt, caries was not recorded as present.”

and in the discussion:

“Therefore, only manifest caries lesions could be determined.”

Reviewer 4 comment: You asked the children about their intake of sweets, sweetened tea and sugary soft drinks. It would be valuable for the reader to get some information about the availability of these products and also about the cost.

Our answer/modification: In the short time on site we were not able to inquire about the cost of sweets in the region. These can vary greatly in Tanzania.

Reviewer 4 comment: You write in the results section that 65.9% of the children had poor oral hygiene- Please elaborate on how this was handled. Poor oral hygiene is a risk factor not only for caries but also for periodontal diseases. Were the children given information about the importance of brushing their teeth and instructions on how to brush their teeth? What was the reason for poor oral hygiene? Did they even have a tooth-brush?

Our answer/modification: We have not yet conducted any follow-up examinations or developed a specific caries prevention program for the region. At the moment, we have recorded the status quo. Parents and children have been informed about the need for treatment. In further investigations we will try to get more information here to develop if necessary meaningful ideas the Kariesprophylaxe. Due to the low prevalence, it may have to come down to a very individual conception. Our vote was limited to the examination of the children. The option of treatment was not allowed. The parents, teachers and students were informed about this in advance. We informed the parents of the affected children about the examination results. We did not make any changes in the text.

Reviewer 4 comment: The consumption of products with a high concentration of easily fermentable carbohydrates was high. Were the children given any information or advice regarding diet? Please also consider adding some information about what other food-items an ordinary diet for Tanzanian children consists of. Despite low dental attendance, poor oral hygiene, not using toothpaste, and a high intake of easily fermentable carbohydrates, the caries prevalence was lower than what might have been expected. What about fluoride in the drinking water? The authors should elaborate more about possible explanations to the caries prevalence.

Our answer/modification: Thank you for the suggestions. These aspects all make sense. We had not included them in the study. However, we plan to do so for future investigations in the region. We have added the following sentence:

“In future studies, the focus should be more on a detailed dietary history and the questioning of previous caries experience. Furthermore, it would be of interest to include more precise information on the fluoride content of the toothpastes used. These aspects would be a useful addition to identify more precisely the causes of caries development in the region.”

Reviewer 4 comment: Line 328. Please describe the oral health education in schools. What ages of the children? Who is in charge of the education?

Conclusion: it would be nice if the authors could add some suggestions on how to improve tooth-brushing frequency and reduced intake of easily fermentable carbohydrates and thereby reduce the risk of caries. The authors should also consider including a line about what you they about the need for dental interventions and prophylactic measures for these children.

Our answer/modification: Thank you for the suggestions. We have not yet conducted any follow-up examinations or developed a specific caries prevention program for the region. At the moment, we have recorded the status quo. Parents and children have been informed about the need for treatment. In further investigations we will try to get more information here to develop if necessary meaningful ideas for caries prophylaxis. Due to the low prevalence, it may have to come down to a very individual conception. Our vote was limited to the examination of the children. The option of treatment was not allowed. The parents, teachers and students were informed about this in advance. We informed the parents of the affected children about the examination results. In the short time on site it was not possible for us to develop a treatment or prophylaxis concept. We are currently examining what possibilities exist for conveying information via the schools. We did not make any changes in the text.